# Cushing’s Myopathy in Dogs: Prevalence, Clinical Abnormalities, and Response to Treatment

**DOI:** 10.3390/ani14213109

**Published:** 2024-10-29

**Authors:** Kate Hovious, Kayla Fowler, Michaela Beasly, Theresa E. Pancotto

**Affiliations:** 1Animal Emergency and Specialty Center, Parker, CO 80134, USA; 2Pinnacle Veterinary Specialists, Glen Mills, PA 19342, USA; kaylamuir22@gmail.com; 3Department of Clinical Sciences, College of Veterinary Medicine, Mississippi State University, Starkville, MS 39759, USA; beasley@cvm.msstate.edu; 4Specialists in Companion Animal Neurology, Naples, FL 34135, USA; tpancotto@scanfl.com

**Keywords:** dog, hyperadrenocorticism, Cushing’s disease, myotonia, pseudomyotonia

## Abstract

Muscular complications of metabolic diseases such as Cushing’s disease can have consequences for the animals they affect. We retrospectively evaluated canine patients with muscular disease secondary to Cushing’s disease along with those reported historically to determine the prevalence of these complications and which treatment strategies are necessary to improve quality of life. We found that while these complications are extremely rare, they do severely impact quality of life in those affected. Unfortunately, once animals are diagnosed, there seems to be little that can be done to reverse the clinical changes. We suspect that prevention and early recognition rather than treatment may be the only way to mitigate the devastating outcomes.

## 1. Introduction

Myotonia is the terminology used to describe the clinical sign of various movement disorders characterized by a disturbance in muscle relaxation after voluntary contraction, percussion, or needle insertion, which typically improves with continued activity, and frequently results in clinically appreciable muscle hypertrophy and difficulty with joint flexion [1,2]. Myotonic discharges on electromyography (EMG) consist of repetitive, high-frequency (20–150 Hz), and high-amplitude (10 mcv to 1 mv) discharges; these discharges acoustically create a characteristic “dive-bomber” sound associated with waxing and waning frequency in the higher range [1,3,4]. Myotonic discharges are not blocked by neuromuscular blocking agents, indicating a primary muscular disorder as opposed to a primary neurogenic disorder [1,2].

The musculoskeletal pathology seen in dogs with hyperadrenocorticism (HAC), or Cushing’s disease, is most often atrophic in nature, frequently causing clinical muscle weakness [5]. In rare cases, HAC can cause muscle stiffness and increased muscle tone [6]. This rare form of myotonia associated with HAC is accompanied by electrophysiologic changes consistent with myotonic or pseudomyotonic discharges [1,4]. Pseudomyotonic discharges are differentiated from myotonic discharges in that the bizarre high-frequency discharges begin and end abruptly, and do not wax and wane [1,4,7].

Electrophysiologic changes associated with hyperadrenocorticism in dogs were first described in the 1970s [8]. In the past 50 years, there have been minimal published data on the topic. A total of 10 cases of dogs with HAC-associated muscle stiffness were identified from four different sources [4,6,8,9] and 37 cases of dogs from a single source [10].

The goal of this study was to retrospectively evaluate a larger population of dogs with hyperadrenocorticism, determine the prevalence of clinical myotonia/pseudomyotonia, and describe the diagnostic evaluation and response to treatment. We hypothesized that HAC-associated myotonia and/or pseudomyotonia would be rare.

## 2. Materials and Methods

Medical records at the Auburn University Small Animal Teaching Hospital, Mississippi State University College of Veterinary Medicine Animal Health Center, and Virginia Maryland College of Veterinary Medicine from 2010 to 2021 were searched for dogs with adrenal disease and myopathy. The inclusion criteria were EMG results consistent with myotonic or pseudomyotonic discharges and diagnosis of hyperadrenocorticism (clinically and/or biochemically). Muscle biopsy results were included when available but were not required for study inclusion. The following data were recorded: patient signalment, diagnosis, weight, age of onset of endocrinopathy-associated clinical signs, age of onset of myotonia, difference in age between myotonia onset and endocrinopathy onset, patient history, method of diagnosis of HAC, progression of signs, evidence of clinical signs of pain, medications at presentation, physical exam findings, neurologic exam findings and neuroanatomic localization, complete blood count, serum chemistry, urinalysis, imaging performed and results, EMG results, muscle +/− nerve biopsy results, treatment initiated, and response to treatment.

### Statistical Analysis

Prevalence was calculated by determining the number of cases of HAC myopathy divided by the number of newly diagnosed cases of HAC from the period of 2010–2021 at each institution. Descriptive statistical analysis (median, mean, SD, range) was performed on data extracted from medical records.

## 3. Results

Fifteen cases were identified, and five dogs were excluded for lack of clinical or diagnostic evidence of HAC. Three of the five dogs also lacked myotonic discharges.

The diagnosis of pituitary-dependent HAC was made by a low-dose dexamethasone suppression test (LDDST) and an ACTH-stimulation test in one case, a single ACTH-stimulation test in two cases, a single LDDST test in two cases, a presumptive diagnosis based on clinical signs, chemistry results, and abdominal ultrasound in one case, and a presumptive diagnosis based on clinical signs and chemistry alone in one case. The method of diagnosis of hypoadrenocorticism in one case was unknown. This patient was included because of steroid administration and concurrent myotonia. No cases of adrenal-dependent HAC were identified in this study.

At institution 1 (AU) there were 351 new cases of HAC recorded from 2010 to 2021 and five cases of HAC with myotonia/pseduomyotonia, with a prevalence of 1.4%. At institution 2 (VMCVM), there were 427 new cases of HAC from 2010 to 2021 and two cases of HAC with myotonia/pseudomyotonia, with a prevalence of 0.46%. At institution 3 (MS), there were 352 new cases of HAC from 2010 to 2021 and one case of HAC with myotonia/pseudomyotonia, with a prevalence of 0.28%. The average prevalence of HAC with myotonia/pseudomyotonia for the three institutions was 0.71%.

There were five spayed females and three neutered males. There were no intact dogs. The weights ranged from 3.62 kg to 15.3 kg, with a median weight of 10.45 kg, a mean weight of 9.9 kg, and a standard deviation of 4.39 kg.

The age range of endocrine disease onset was 3–10 years old, with a mean age of 7.8 years, a median of 8.5 years, and a standard deviation of 3.8 years. The age range of myotonia onset was 4–13 years old, with a mean age of 8.9, a median of 9.25 years, and a standard deviation of 3.5 years. The difference between age of onset of myotonia symptoms and age at endocrine diagnosis ranged from 0 to 2 years, with a mean of 1.05 years, a median of 1 year, and a standard deviation of 0.875 years.

The progression of myotonia symptoms was chronic and progressive in six cases, chronic and static in one case, and acute onset in one case.

Five cases were reportedly painful for the patient or on physical examination, and three cases did not have a pain response noted in the medical record.

The patient exam data are summarized in Table 1. On physical exam, all the dogs had skin changes, including alopecia (six out of eight), thin skin (four out of eight), hyperpigmentation (one out of eight), comedones (one out of eight), hyperkeratosis (one out of eight), calcinosis cutis (one out of eight), and papules (one out of eight). All the dogs also had muscle changes, including contractures or rigidity in seven out of eight, atrophy in four, and hypertrophy in two. Three dogs (three out of eight) had hepatomegaly and two out of eight had a pendulous abdomen.

On neurologic exam, four out of eight dogs had gait abnormalities, including stiffness (three), bunny-hopping (two), and ataxia (one). Seven out of eight dogs had proprioceptive deficits; two dogs had them in all four limbs, two dogs in the pelvic limbs only, two dogs in the thoracic limbs only, and one dog hemilaterally. Reflexes were decreased in seven out of eight dogs, five of which were associated with increased muscle tone and inability to elicit reflexes. Generalized muscle weakness was noted in two out of eight dogs and one dog had muscle tremors noted in the thoracic limbs. No mentation changes or cranial nerve deficits were noted. In all eight cases, neurolocalization was to the motor unit, with two cases having concurrent spinal cord involvement, one cervical involvement, and one thoracolumbar involvement.

Complete blood count was performed in seven out of eight cases. The CBC was normal in two out of seven dogs, three out of seven dogs had leukocytosis, three out of seven dogs had neutrophilia, three out of seven dogs had lymphopenia, one out of seven dogs had eosinopenia, five out of seven dogs had thrombocytosis, and one out of the seven dogs had anemia.

Serum biochemistry was performed in seven out of eight cases. An increased ALP was noted in six out of seven dogs, four out of seven dogs had an increased ALT, six out of seven dogs had an increased AST, two out of seven dogs had an increased GGT, four out of seven dogs had an increased CK, two out of seven dogs had hyperalbuminemia, five out of seven dogs had hypercholesterolemia, one dog out of seven had hypocalcemia, one out of the seven dogs had hypercalcemia, one out of the seven dogs had hypertriglyceridemia, and four out of seven dogs had hyperglycemia.

Urine analysis was performed in five out of eight dogs. Urinalysis was normal in two out of the five dogs. Proteinuria was identified in three out of the five dogs. Urine specific gravity was minimally concentrated in two out of the five dogs (1.019, 1.021).

Thoracic radiographs were performed on five out of eight dogs. One case was within normal limits, three cases had hepatomegaly, three cases had degenerative joint disease, and one case had an enlarged proximal descending aorta. Abdominal radiographs were performed in one case, with no significant findings. Abdominal ultrasound was performed in five out of eight cases. All five of these cases had hepatomegaly with a hyperechoic liver, three cases had gallbladder abnormalities (sludge, thickened wall, and polypoid), four cases had kidney abnormalities (cyst, nephrolith, degeneration, and hyperechoic cortical foci), and two had adrenal changes, including one with bilateral enlargement and one with mineralization. Spinal radiographs were obtained in one out of eight cases and showed articular facet arthritis in the thoracolumbar vertebral column. A spinal MRI was performed in two out of eight cases; one case was diagnosed with C5–7 intervertebral disc disease without compression and one case was normal.

All the electrodiagnostic studies were performed by a board-certified veterinary neurologist or neurology resident under the supervision of a board-certified veterinary neurologist. EMG was performed using a concentric needle electrode and NCV was performed using monopolar electrodes.

EMG was performed in eight out of eight cases, with a full report or study available in four out of eight. Four cases had pseudomyotonic discharges. Four cases had complex repetitive discharges. One case had myotonic discharges. One case had a written interpretation that the EMG was consistent with generalized neuromuscular disease, but specific EMG changes were not further described or available for review. Motor nerve conduction velocity was performed in three out of eight dogs, with one normal, one mildly delayed NCV suspected to be age-associated, and one written interpretation that said that the findings were consistent with demyelination, but no studies were available for review.

Muscle biopsies were performed in five out of eight dogs, but only three reports were available. The muscle samples included two dogs with quadriceps only, one dog with biceps femoris and triceps, one dog with semimembranosus only, and one dog with quadriceps and triceps. Dog 1’s histopathology was described as showing moderate variability in myofiber size, with marked type 1 fiber predominance; excessive and large intramyofiber lipid droplets are present within type 1 fibers. Dog 4 showed chronic non-inflammatory myopathy with marked type 2 fiber atrophy; excessive intramyofiber lipid droplets and lobulated fibers consistent with an endocrine disorder. Dog 5 showed chronic, severe non-inflammatory myopathy with type 1 fiber predominance and extensive myofiber lobulation consistent with chronic Cushing’s syndrome and pseudomyotonia. A nerve biopsy was performed in one out of eight dogs using the peroneal nerve, but no report was available.

Cerebrospinal fluid was sampled in two dogs that had elevated total protein (26.7 mg/dL and 43.4 mg/dL), two dogs had a negative urine culture, one dog had an arthrocentesis (elbow, stifle) that showed mild suppurative inflammation, and one dog had a liver fine needle aspirate showing moderate vacuolar hepatopathy.

The infectious disease testing included three dogs with a negative Neospora caninum serology IFA, three dogs with a negative Toxoplasma gondii antibody titer (IgG + IgM), two dogs with negative Hepatozoon PCR, and one dog for each of the following: negative canine distemper virus antibody titer (IgG + IgM), negative Ehrlichia canis antibody titer (IFA), and negative Rocky Mountain spotted fever antibody titer (IFA).

Gabapentin was prescribed in five dogs (6–14 mg/kg BID-TID). Tramadol was prescribed in two cases (3.2–5.5 mg/kg TID). Carprofen 2 mg/kg BID was prescribed in one case. Methocarbamol was prescribed in two cases at doses of 33 mg/kg TID and 22 mg/kg BID. Diazepam 2 mg/kg TID was prescribed in one case. Dantrolene 3.7 mg/kg BID was prescribed in one case. Procainamide was prescribed in three cases (1.4 mg/kg BID, 12.5 mg/kg BID). Phenytoin 30 mg/kg TID was prescribed in one case. Mexiletine was prescribed in one case at 10 mg/kg TID. Vitamin B12 injections at an unknown dose into an acupuncture site on the head were used in one case. Co-enzyme Q10 was prescribed in two cases, one at an unknown dose and the other at a dose of 100 mg/kg SID. Carnitine was prescribed in three cases, one at an unknown dose, one at a dose of 44 mg/kg BID, and one at 68 mg/kg/day. Physical rehabilitation was prescribed in three cases, which improved mobility in one case (laser, underwater treadmill, hot packing, and passive range of motion), but no improvement was recorded in two cases (massage and unknown therapeutic exercises). One of these dogs was fitted for a cart for mobility assistance as part of the physical rehabilitation protocol. Two cases were taking prednisone at the time of diagnosis of myopathy, one with hypoadrenocorticism and the other with historic IMHA, and the treatment consisted of tapering off prednisone. Two dogs were treated with mitotane (30 mg/kg AM, 20 mg/kg PM; 37 mg/kg SID) and one dog was treated with trilostane (dose unknown). Three dogs did not receive HAC-directed treatment. One dog was treated with clindamycin 14.7 mg/kg BID.

Necropsy was performed on one dog that was euthanized due to progression. The necropsy results revealed diffuse CNS granulomatous inflammation, hepatocellular vacuolation consistent with steroid hepatopathy, lymphoplasmacytic enteritis, and pulmonary edema. This patient had previously had a muscle biopsy showing muscle atrophy and fatty replacement with no evidence of inflammation, and a liver fine needle aspirate showing moderate vacuolar hepatopathy.

## 4. Discussion

We identified eight cases of HAC-associated myopathy at three different institutions, with an average prevalence of 0.71%. Six of the dogs were diagnosed with pituitary-dependent HAC, one dog with suspected iatrogenic HAC, and one dog with hypoadrenocorticism and, later, suspected iatrogenic HAC. Iatrogenic HAC was presumed when the patients were receiving oral steroids and had clinical signs of Cushing’s disease, including thin skin, clacinosus cutis, diffuse muscle wasting, high blood pressure, and/or elevated liver enzymes. The last dog did not exhibit clinical signs consistent with iatrogenic HAC but was on oral prednisone, and the clinical pathological results and development of myotonia suggested steroid-associated pathology.

In the previous literature, consisting of 47 dogs in five different reports, 2 cases were diagnosed with iatrogenic HAC, and the other 45 developed HAC secondary to a pituitary adenoma [4,6,8] or macroadenoma [11]. This is consistent with the results of this study, in which six out of the eight dogs that developed myotonia were diagnosed with pituitary dependent HAC. It is interesting that of the dogs that developed myotonic/pseudomyotonic discharges, none were diagnosed with HAC secondary to an adrenal tumor. To our knowledge, a case of myotonia/pseudomyotonia secondary to adrenal-dependent HAC has not been reported in the veterinary literature. In general, amongst dogs diagnosed with HAC, 85% are pituitary-dependent [11]. More data are needed in order to demonstrate the development of myotonia/pseudomyotonia in adrenal-dependent HAC.

In the current study, we identified one case treated for hypoadrenocorticism that went on to develop myotonia, without clinical signs of HAC. This dog’s clinical signs of myotonia/pseudomyotonia occurred years after initiating treatment for hypoadrenocorticism with prednisone and fludrocortisone. Muscle stiffness was first seen in the right hindlimb and progressed to involve the left hindlimb over the course of 5 months. At the time of presentation, the dog was receiving prednisone at 0.3 mg/kg/d and fludrocortisone at 0.01 mg/kg twice daily. Clinical pathology data showed mild/moderate leukocytosis with high ALP (581), ALT (164), AST (78), CK (907), and hyperglycemia (120). The electrolytes were within normal limits. A urinalysis was not performed. Further endocrine testing was not performed. The EMG findings were consistent with pseudomyotonic discharges, and the muscle biopsy results were consistent with metabolic myopathy. The treatment consisted of decreasing the prednisone dose to 0.25 mg/kg/d, discontinuing fludrocortisone and replacing it with monthly DOCP injections, mexiletine at 10 mg/kg TID, methocarbamol at 33 mg/kg TID, gabapentin 7–14 mg/kg TID, carnitine at 1000 mg daily, and physical therapy. The treatment was unsuccessful at improving the clinical symptoms.

Three dogs in this study (37.5%) had concurrent hypothyroidism and were being treated with thyroxine at the time of diagnosis with myotonia. Both chronic hypothyroidism and HAC in dogs have been associated with type 1 fiber predominance and type 2 fiber atrophy [12,13]. This is also seen in humans with hypothyroidism and glucocorticoid excess [14]. In humans with hypothyroidism, the clinical signs include enlarged muscles, muscular weakness, and delayed muscle relaxation that do not improve with repetition, differentiating this clinical sign from true myotonia [15]. To our knowledge, there are no reports of canine hypothyroidism causing myotonia-like symptoms in affected dogs. All the dogs in this study with HAC and hypothyroidism were being treated with thyroxine at the time of diagnosis with myotonia. This warrants the question of whether there may be a multi-factorial pathophysiological mechanism in which the combination of hyperadrenocorticism and hypothyroidism causes an increased risk of developing myotonia-like symptoms compared to dogs with HAC alone. Of the forty-seven cases in the literature, four (8.5%) had been concurrently diagnosed with hypothyroidism [4,6,8,9,10]. The significance of these data is unclear and warrants further investigation with a larger number of subjects.

In our study, all the dogs had a chronic history of HAC, HAC-clinical signs, and/or HAC-clinicopathologic abnormalities. The average duration between onset of HAC and diagnosis of myotonia was 12.6 months. Four of the dogs (37.5%) were undergoing treatment for HAC at the time of evaluation for myotonia. In the 47 cases from the literature, 24 of the cases (51%) were previously diagnosed with and were undergoing treatment for HAC prior to the onset of myotonia signs. Five of the forty-seven dogs from the literature review were diagnosed with hyperadrenocorticism at the time that they were first seen for myotonia, one had an unknown history, and one was diagnosed with HAC at necropsy [4,6,8,9,10].

Physical exam changes consistent with poorly controlled HAC, particularly dermatologic changes, were seen in all the cases evaluated for the present study. This is consistent with cases reported in the literature [4,6,8,9,10]. These findings may suggest that the development of myotonia is more likely in patients whose Cushing’s disease is poorly controlled. This is an area for further investigation. The most common neurologic clinical signs in our study included proprioceptive deficits (seven out of eight), decreased reflexes (seven out of eight), and a bunny-hopping gait due to muscle stiffness (three out of eight). Five of the dogs with decreased reflexes were suspected to have these changes secondary to severe muscle stiffness and loss of range of motion. Of the cases in the literature with a neurologic exam report, six of the ten dogs had no neurologic changes other than those attributable to muscle rigidity. This suggests neurolocalization to the neuromuscular junction.

Seven cases had a CBC and chemistry available. The most common CBC abnormalities included thrombocytosis (five out of seven) and a stress leukogram (two out of seven). The most common chemistry abnormalities included elevations in ALP (six out of seven), AST (six out of seven), ALT (four out of seven), cholesterol (four out of seven), CK (4/7), and blood glucose (four out of seven). Urinalysis was available in five dogs; proteinuria was the only commonality and was present in three out of five cases. Not unsurprisingly, these findings are all consistent with a diagnosis of HAC. This is similar to cases in the previous literature and further supports the hypothesis that dogs with uncontrolled or poorly controlled HAC are more likely to develop myotonia [4,6,8,9,10].

Although the EMG descriptions were abnormal in eight out of the eight dogs, full EMG studies were only available for review in four cases. The most common EMG findings included pseudomyotonic discharges (four out of eight) and complex repetitive discharges (four out of eight). Only one dog showed myotonic discharges. Similarly, the literature documents bizarre, high-frequency discharges, occasionally described as having a “dive-bomber” quality, with occasional myotonic discharges [4,6,8,9,10].

Muscle biopsies were performed in five out of eight dogs. The most common findings were selective type 2 fiber muscle atrophy, marked type 1 fiber predominance, and replacement of myofibers by adipose tissue, consistent with non-inflammatory endocrine myopathy. This is in agreement with historical cases of HAC-associated myopathy [6,8,10].

The treatment was highly variable even within this small population. The most common therapeutic strategy was treatment with analgesics, used in seven out of eight cases. Weaning steroids or treating for HAC was recorded for five out of eight dogs. Muscle relaxants, sodium channel blockers, and supplements were each used in four dogs. Six out of eight dogs did not show significant improvement. One dog remained static, and one was lost to follow-up. Nine of the forty-seven dogs in the previous literature showed mild to marked improvement with various combinations of the therapies utilized in the present study; however, the vast majority of the cases in the literature did not improve or had progressive disease [4,6,8,9,10]. We are unable to make any useful conclusions about the best underlying treatment strategy. Treatment of the underlying HAC is recommended despite the lack of improvement specifically seen in myotonia-associated symptoms in this study. Although Cushing’s myopathy is uncommon, there is a need for better treatment of this condition. In some patients, the myopathic consequences caused significant detrimental changes to quality of life, resulting in euthanasia in one dog [8] and the need for a wheelchair in another. Early diagnosis and treatment of HAC or the removal of prolonged steroids may be the most important factors for prevention given the poor response to treatment.

## 5. Conclusions 

There are several weaknesses in our study, including the small sample population. This is despite contributions from multiple institutions. Several institutions were unable to contribute cases due to incomplete medical records. HAC was presumptive in some patients and endogenous ACTH measurements were not available for any cases. Not all the cases had EMG tracings that could be reviewed. We included cases as long as a report was available in order to not further reduce the population. Similarly, biopsies and biopsy reports were only performed in five out of eight dogs. The treatment strategies were inconsistent, and we are unable to make any conclusions about whether HAC with myotonia/pseudomyotonia is truly refractory given the lack of knowledge about treatment. Future studies should concentrate on more consistent evaluation of a particular therapeutic strategy.

## Figures and Tables

**Table 1 animals-14-03109-t001:** Physical exam findings.

Physical Exam Changes	# of Dogs (%)
Dermatologic (alopecia, thinning skin, hyperpigmentation, comedones, hyperkeratosis, calcinosis cutis, papules)	8/8 (100)
Muscle contracture/rigidity	7/8 (87.5%)
Muscle atrophy	4/8 (50%)
Hepatomegaly	3/8 (37.5%)
Muscle hypertrophy	2/8 (25%)
Pendulous abdomen	2/8 (25%)

## Data Availability

Data is contained within the article.

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
