# Peer review of "Cushing’s Myopathy in Dogs: Prevalence, Clinical Abnormalities, and Response to Treatment"

_animals, 2024, doi:10.3390/ani14213109_

Round 1
Reviewer 1 Report
Comments and Suggestions for Authors
The authors are describing the Cushing myopathy in a population of 8 dogs. The topic is interesting and might be usefull for both the general practitioners and veterinary neurologists.
However, the lack of group homogenity creates difficulties in interpreting the results but also in understanting if the researsch goals were achieved . I consider the MM section should be improved (if posible). In the actual form the EDX exam is difficult to be followed (also a good part of the patients have incomplete to absent / not homogenous findings). Maybe will be usefull to see in a table the results of edx exams in all dogs.
l 113 - please elaborate the neurological exam
l 115 - which muscles?
l 140 - i consider the edx methodology should be added.
l 145 - the edx exam is vague. Again, maybe will be easier to be followed by putting the results in a table.
l 148 - for the histo report: please describe de findings
I believe by improoving the MM section (especially the neuroexam and edx findings) will be usefull in understanding a specific muscle group mostly affected)
Author Response
Dear Reviewers,
We thank you for these thoughtful and constructive comments. We have done our best to incorporate all suggestions. Given the age of some of the records and variability between institutions, there are limitations to the detail of the medical record and consistencies in technique. However, we felt it was still important to include as many cases as possible to reflect the rarity of this disease.

Reviewer 2 Report
Comments and Suggestions for Authors
General comments
The manuscript presents a study on Cushing’s Myopathy. The authors have conducted extensive research and provided valuable insights. However, there are several areas that need substantial improvement before the manuscript can be considered for publication. There are major concerns that need to be addressed.
The literature review is out of date. The authors have missed several key studies crucial for understanding the context and significance of their work. I recommend expanding the literature and citing more recent work on the topic, such as:
- Golinelli S, Fracassi F, Bianchi E, et al. Clinical features of muscle stiffness in 37 dogs with concurrent naturally occurring hypercortisolism. J Vet Intern Med. 2023;37(2):578‐585. doi:10.1111/jvim.16620
- Burbaitė E, Fiorentino E, Negro L, Menchetti M. Stiffness of the four limbs in a Jack Russell Terrier dog. J Am Vet Med Assoc. 2023 Oct 16;262(1):139-141. doi: 10.2460/javma.23.07.0424. PMID: 38103386.
The materials and methods section lacks sufficient details, making it difficult to assess the validity of the design of the study. The authors should provide a more detailed description of the methods used; please better define the inclusion criteria for the diagnosis of hyperadrenocorticism, specifying what endocrine tests were performed. Please include only patients with a confirmed diagnosis of HAC.
The tables are not clearly labeled or are missing critical information. Ensure references to tables in the text are correct and complete.
Addressing these major revisions will significantly enhance the quality of the manuscript and its potential for publication.
I look forward to reviewing a revised version of this work.
Specific comments
Line 1: “Case Report”. The authors aim to assess the prevalence of this rare condition and retrospectively evaluated cases of hyperadrenocorticism diagnosed at three centers. It would be more correct to call it a retrospective crossover observational study
Line 19: EMG, write in full the first time
Line 22: in the abstract is defined as a retrospective study, keep this definition
Line 37: layout/formatting error
Lines 52-53: The literature is out of date. I recommend expanding the literature and citing more recent work on the topic, for example:
- Golinelli S, Fracassi F, Bianchi E, et al. Clinical features of muscle stiffness in 37 dogs with concurrent naturally occurring hypercortisolism. J Vet Intern Med. 2023;37(2):578‐585. doi:10.1111/jvim.16620
- Burbaitė E, Fiorentino E, Negro L, Menchetti M. Stiffness of the four limbs in a Jack Russell Terrier dog. J Am Vet Med Assoc. 2023 Oct 16;262(1):139-141. doi: 10.2460/javma.23.07.0424. PMID: 38103386
Line 55: “Incidence”. In the title the authors talk about the prevalence of the disease
Line 62-63: “EMG results consistent with myotonic or pseudomyotonic discharges”. In which patients has electromyography been done? On what basis was Cushing's myopathy suspected?
Lines 63-64: “evidence of hyperadrenocorticism (clinically and/or biochemically)”. Please better define the inclusion criteria for the diagnosis of hyperadrenocorticism. What tests have been performed?
The authors' goals include describing the prevalence of the condition; it is important to strictly define the inclusion criteria and include only those patients who meet them.
In the results, the authors mention pituitary-dependent and adrenal-dependent HAC; on what basis were the two forms differentiated?
In the results, the authors mention iatrogenic HAC. Please clarify how is made the diagnosis of this condition
Line 72: “complete blood count, serum chemistry, urinalysis, imaging performed and results”. Were these tests performed at the time of diagnosis? Please specify
Please describe in detail the equipment used and the procedure performed
Lines 80-85: Please consider only patients who have a confirmed diagnosis of HAC
Lines 86-88: given the number of patients included, it is recommended to produce a table that contains specifics on diagnosis and treatment of each dog
Line 89: since HAC is a condition diagnosed in adult/older patients, it is recommended to use “years” instead of “months”
Lines 97-98: the authors are requested to include only patients who have a confirmed diagnosis of HAC
Line 106: there is no reference to the table 1 in the text
Line 138: IVDD, write in full the first time
Line 154: layout/formatting error
Line 108: there is no reference to the table 2 in the text
Line 192: please include only cases with confirmed diagnosis of HAC. Please clarify how is made the diagnosis of iatrogenic HAC
Lines 198-200: The literature is out of date. I recommend expanding the literature and citing more recent work on the topic, for example:
- Golinelli S, Fracassi F, Bianchi E, et al. Clinical features of muscle stiffness in 37 dogs with concurrent naturally occurring hypercortisolism. J Vet Intern Med. 2023;37(2):578‐585. doi:10.1111/jvim.16620
- Burbaitė E, Fiorentino E, Negro L, Menchetti M. Stiffness of the four limbs in a Jack Russell Terrier dog. J Am Vet Med Assoc. 2023 Oct 16;262(1):139-141. doi: 10.2460/javma.23.07.0424. PMID: 38103386.
Lines 208-210: “In the current study we identified one case treated for hypoadrenocorticism that went on to develop myotonia, without clinical signs of HAC”. The diagnosis of this case is doubtful; the authors are requested not to include it in the study if the diagnosis is not supported by clinical and endocrine tests that confirm the diagnostic suspicion. Have been performed other investigations to rule out any concomitant diseases?
Lines 223-224. Please clarify how the diagnosis of hypothyroidism was made in these patients. Was it performed before the diagnosis of HAC?
Line 228: layout/formatting error
Line 229: “To our knowledge at the time of writing this paper”. Redundant sentence
Line 236-237: The literature is out of date. I recommend expanding the literature and citing more recent work on the topic
Lines 239-299: Please review the discussion in light of the updated literature
Line 306. another limitation is the lack of measurement of endogenous ACTH in the population included; eACTH is useful to discriminate between pituitary-dependent and adrenal-dependent HAC and accurately diagnose an iatrogenic HAC.
Author Response

(The authors gave the same response as above.)

Reviewer 3 Report
Comments and Suggestions for Authors
1. "EMG" in Abstract and the second paragraph of Introduction is for electromyography? Abbreviations should be defined with their full names when they first appear, please check other abbreviations.
2. Keep a space between the numbers and their units, e.g. Line 86-88, check this issue throughout the manuscript.
3. There are many very short paragraphs, it's recommended to re-organize these paragraphs.
4. Table 1 and Table 2 are not very informative, it's kind of waste of space.
5. Since this report is about canine patients/dogs, it's better to have "canine or dog" in the title.
This case report discussed a uncommon disease Cushing’s Myopathy in dogs. The contents including the Prevalence, Clinical Abnormalities, and the Response to treatment, based on relatively small sample population. Since it's uncommon, there is only a few related literature about this disease. So the present retrospective evalation report is quite important for the better understanding and treatment to this disease in veterinary medicine. The manuscript has included a paragraph about their weakness, I totally agree with it. My only extra comments is that this report needs to be reorganized for some very short paragraphs and the two tables.
Comments on the Quality of English Language
Basically, the quality of English expression is good.
Author Response

(The authors gave the same response as above.)

Reviewer 4 Report
Comments and Suggestions for Authors
This retrospective study aims to evaluate dogs with hypercortisolism (iatrogenic or spontaneous) and myotonia. The topic is very interesting, as are the data derived from it. However, the article is quite confusing and does not adhere to the guidelines for the definition of hypercortisolism, Cushing's syndrome, and their diagnosis. Furthermore, some data are incorrectly reported multiple times, and the discussion section is merely a repetition and elaboration of the results without a genuine analysis of them.
I suggest revising the article with the provided suggestions, focusing on the accurate diagnosis of the included cases and the discussion of the obtained data.
Cushing’s disease refers to pituitary-dependent hypercortisolism in humans and is not a term frequently used in veterinary medicine. I suggest the authors use the ALIVE definition of the ESVE (European Society of Veterinary Internal Medicine) for spontaneous hypercortisolism or Cushing’s syndrome.
Line 43: Use the term "hypercortisolism" instead of "hyperadrenocorticism" and "Cushing’s syndrome" instead of "Cushing’s disease." This must be corrected throughout the paper.
Line 66: What were the inclusion criteria for the diagnosis of Cushing’s syndrome? What clinical signs? What biochemical abnormalities? What tests? How did you differentiate between ADH and PDH?
Line 97: To include a dog with a presumptive diagnosis of Cushing’s syndrome, the authors must describe the single case: what were the clinical signs, and what were the blood exam abnormalities? Otherwise, these cases must be excluded.
Line 99: Hypoadrenocorticism?
Line 179: You must be clearer in the inclusion criteria. Did you also include dogs with iatrogenic hypercortisolism? In line 61, you mentioned adrenal disease without specifying which one, and this is very important. How did you perform the diagnosis of Cushing’s syndrome in the dog on prednisone for IMHA? Was this a diagnosis of iatrogenic hypercortisolism? All this information is missing from the paper.
Line 189: You wrote that 8/8 dogs underwent EGM. How is it possible that this dog did not have the EGM ante-mortem? Was EGM not an inclusion criterion of your study?
Line 226: I would avoid the part on hypothyroidism. Were you sure about the diagnosis? Did all the dogs show an increased TSH and low T4? If yes, you can keep this part. If not, I would delete this paragraph. Many Cushing’s dogs show a low T4 due to the effect of cortisol and not because they are hypothyroid. The myotonia is probably due to hypercortisolism and not hypothyroidism in these dogs.
Line 247: I would modify this part. In many of the myotonia case reports, dogs were under Cushing’s treatment, and nothing changed. So, I would not speculate that treating the hypercortisolemic state would improve the myotonia.
Line 269: This data does not correspond with the data reported in line 122, where 7/7 dogs had an increase in AST. Please check all your data again.
Author Response

(The authors gave the same response as above.)
